# Population Genetic Structure of Citrus Tatter Leaf Virus in Zhejiang Province, China

**DOI:** 10.3390/v17070909

**Published:** 2025-06-27

**Authors:** Lianming Lu, Shunmin Liu, Zhanxu Pu, Baoju An, Danchao Du, Xiurong Hu, Jia Lv, Zhendong Huang

**Affiliations:** Zhejiang Citrus Research Institute, Zhejiang Academy of Agricultural Sciences, Taizhou 318020, China; minglu79@126.com (L.L.); 13044588096@163.com (S.L.); puzhanxu@139.com (Z.P.); anbj2015@163.com (B.A.); dcdu84@163.com (D.D.); 13906573967@163.com (X.H.); jiascolytidae@126.com (J.L.)

**Keywords:** citrus tatter leaf virus, genetic diversity, population structure, host-driven adaptation

## Abstract

Citrus tatter leaf virus (CTLV), a major pathogen threatening global citrus production, remains poorly characterized in terms of its regional genetic diversity and evolutionary dynamics. To address this gap, we conducted a comprehensive population genetic analysis of CTLV in Zhejiang Province, China, using 181 coat protein (CP) gene sequences—the largest regional CTLV dataset to date. Our analyses uncovered substantial genetic diversity among Zhejiang CTLV isolates. Phylogenetic reconstructions revealed that these isolates span multiple clades, closely aligning with global CTLV population structures, indicative of recurrent viral introductions and extensive regional circulation. Population structure analyses revealed significant genetic differentiation driven by geography, with Jinhua isolates forming a distinct cluster, and by host species, with *Citrus reticulata* ‘Criton’ isolates diverging from those in other citrus varieties. Selection pressure analysis indicated that while most CP polymorphic sites were under purifying selection, several clade-specific codons showed signatures of positive selection. These results offer new insights into CTLV’s population structure and localized evolutionary trajectories, enhancing our understanding of its regional adaptation and informing strategies for disease management and control of this globally significant pathogen.

## 1. Introduction

Citrus is one of the world’s most widely cultivated fruit crops, grown in over 140 countries, primarily in tropical and subtropical regions [1]. Citrus production is widespread throughout southern China, with total output reaching approximately 64.34 million metric tons in 2023 (http://faostat.fao.org, accessed on 20 June 2024), making China the world’s leading citrus producer [2]. However, citrus tatter leaf disease (CTLD) is a serious viral disease that poses a significant threat to citrus yield and quality in China. The causal pathogen, citrus tatter leaf virus (CTLV), was first identified in California, USA, as a latent infection in Meyer lemon trees [*Citrus limon* (L.) Burm. f. hyb.] [3]. Since then, CTLV has been detected in Australia, Japan, South Africa and China [4].

CTLV is currently considered a strain of apple stem grooving virus (ASGV), a member of the genus *Capillovirus*, in the family *Betaflexiviridae* [5]. It is transmitted mechanically among citrus plants. While most citrus cultivars serve as latent hosts for the virus, CTLV infection in cultivars grafted onto trifoliate orange or its hybrids often results in stunted growth, chlorosis and virus-induced bud union incompatibility [6]. Beyond citrus, CTLV has been documented to infect a variety of herbaceous plant hosts, the majority of which remain symptomless, further complicating detection and management strategies. Like other capilloviruses, CTLV has a single-stranded RNA genome of approximately 6500 nucleotides, containing two overlapping open reading frames (ORFs) [5]. ORF1 encodes a polyprotein containing a replication-associated protein fused with a coat protein (CP), while ORF2 encodes the movement protein (MP). The CP is expressed from a subgenomic RNA and is thought to play an important role in viral infection [7].

Zhejiang Province, located in southeastern China, is an important production region for citrus. The main citrus-growing areas in Zhejiang are distributed among the cities of Taizhou (35%), Quzhou (25%), Ningbo (16%) and Lishui (13%) [8]. However, CTLV remains a significant threat to the Zhejiang citrus industry, where trifoliate hybrids have become the dominant rootstocks recently. CTLD in Zhejiang was first identified in the 1970s by Japanese plant quarantine authorities on several citrus varieties imported from Huangyan City. By the mid-1980s, Zhang et al. [9] confirmed CTLD infection in 10 citrus varieties cultivated in Huangyan. In the early 1990s, Xie et al. [10] subsequently detected this disease in eight additional varieties grown in Wenzhou City.

Studies on the population genetic structure of plant viruses are essential for understanding their evolutionary history, including demographic history and adaptative evolution [11]. Such analyses elucidate the mechanisms driving genetic differentiation within and between regions, thereby informing more effective disease management strategies, clarifying virus–host interactions and providing robust estimates of population parameters for epidemiological modeling. Although Shokri et al. [12] conducted the first global-scale analysis of ASGV population genetics, their study was limited by sparse and geographically scattered sampling. This leaves significant gaps in our understanding of CTLV genetic diversity, particularly with respect to (1) variation across different host species and (2) population structure at finer geographic scales. To fill these critical gaps, we performed an in-depth analysis of CP gene sequences of 181 CTLV CP gene sequences from Zhejiang Province, including 84 newly obtained isolates. This study had two main objectives: first, to characterize the fine-scale population genetic structure of CTLV in this key citrus-growing region and, second, to identify the evolutionary forces, such as selection pressures and demographic processes, shaping the observed genetic diversity.

## 2. Materials and Methods

A total of 181 CTLV isolates were analyzed in this study, 84 of which were newly obtained, while the remaining isolates had been reported in our previous work [13]. The 181 isolates were randomly collected from 59 sampling locations across Zhejiang Province during 2019 and 2020 (Figure 1, Appendix A). These isolates were classified into seven distinct populations based on their geographical origin: Jiaxing (*n* = 4), Jinhua (*n* = 12), Lishui (*n* = 18), Ningbo (*n* = 40), Quzhou (*n* = 4), Taizhou (*n* = 75), Wenzhou (*n* = 25) and Zhoushan (n = 3). The viral isolates were obtained from various citrus species and hybrids, including Criton (*n* = 8), citrus hybrid (*n* = 138), mandarin (*n* = 26), orange (*n* = 2), pummelo (*n* = 4) and tangerine (*n* = 3), representing diverse types and interspecific hybrids within the *Citrus* genus.

To sequence the CP sequences of CTLV, total RNAs were extracted using an RNA Kit AP-MN-MS-RNA-250 according to the manufacturer’s instructions (Axygen, Hangzhou, China). One-step PCR amplifications were performed in a total volume of 50 μL composed of 25 μL of 2 ×One Step Mix, 0.5 μL of forward primer (10 μmol/L), 0.5 μL of reverse primer (10 μmol/L), 20.5 μL of ddH_2_O and 1.0 μL of template cDNA. The thermal profile included a reverse transcription at 50 °C for 30 min, an initial denaturation at 94 °C for 3 min, followed by 35 cycles of 94 °C for 30 s, 57 °C for 30 s and 72 °C for 1 min, with a 10 min extension at 72 °C. Following the completion of PCR, the amplified product was recovered and ligated into the T-tailed pEASY-T5 cloning vector (TransGen, Nanjing, China). Blue-white screening was then carried out to identify successful clones, after which the recombinant plasmid was extracted and sequenced in both directions using M13 primers by Sangon Biological Co., Ltd. (Shanghai, China). At least three cDNA clones from each transformation were sequenced to obtain a consensus sequence. Sequences were assembled using DNAMAN 12.0 (Lynnon, Quebec, Canada).

Nucleotide sequences of the CP gene of CTLV were aligned using the codon-based model with the MAFFT algorithm [14], implemented in PhyloSuite 1.2.3 [15]. Haplotype diversity (*H*_d_) and nucleotide diversity (*π*) were calculated using DnaSP 6.0 [16]. Pairwise *F*_ST_, a measure of genetic differentiation among populations, was computed in Arlequin 3.5 [17]. The hypothesis of deviation from null population differentiation was tested through 1000 permutations of the original data. Simultaneously, the *F*_ST_ value was also used to assess gene flow between populations. Following standard interpretation thresholds, *F*_ST_ values below 0.33 suggest frequent gene flow, while values above 0.33 indicate more restricted gene flow [18]. In addition, calculations of population differentiation based on *G*_ST_ were performed using the PopGenome 2.16 R package [19]. We also performed discriminant analysis of principal components (DAPC) to investigate the genetic structure of CTLV populations using the adegenet 2.1.11 R package based on predefined groups [20]. This multivariate approach partitions sample variance into between-group and within-group components, maximizing discrimination among groups. Unlike some other methods, DAPC does not assume panmixia.

To examine the potential geographic and host-origin effects on CTLV CP diversification, we performed a phylogeny–trait association analysis using BaTS 2.0 [21]. The strength of association between phylogeny and traits was measured through the calculation of three statistical methods to assess association index (*AI*), parsimony score (*PS*) and maximum monophyletic clade (*MC*), and the comparison of their values with the ones computed from 1000 location-randomized trees. *p* < 0.05 was considered significant, indicating a strong phylogeny–trait association.

To further investigate specific sites for episodic adaptive evolutionary selection in the CTLV, two different approaches were employed. First, we calculated the non-synonymous-to-synonymous substitutions ratio (*ω* = *d*N/*d*S) using the codon-based ML method in EasyCodeML 1.41 [22], a user-friendly alternative to CodeML that provides visualized operations. The site model was used to detect positively selected sites within the CP [23]. Three nested models (M3 vs. M0, M2a vs. M1a and M8 vs. M7) were compared, and likelihood-ratio tests (LRTs) were conducted to determine the best fit for the data. When the LRT results were significant (*p*-value < 0.05), the Bayes empirical Bayes (BEB) method was employed to identify amino acid residues likely to occur under positive selection [24]. In addition, *ω* values across the site were extracted from the M3 model results for the generation of sliding window plots. Second, we employed the mixed-effects model of evolution [25] implemented in the Datamonkey server [26]. In this analysis, MEME estimates two classes of ω values along with their corresponding weights: under the null model, the *α* value (*d*S) is unconstrained, while the two separate *β* values (*d*N; *β*^+^ and *β*^−^) are both constrained, with *β^+^* ≤ *α* and *β*^−^ ≤ *α*. Under the alternative model, only *β*^−^ is constrained (*β*^−^ ≤ *α*), whereas *α* and *β*^+^ remain unconstrained. If the LRT between these two models is statistically significant and *β*^+^ > *α*, the inferred site is under positive selection.

To explore the demographical history of the CTLV population, Bayesian skyline plots (BSP), which estimate changes in effective population size through time, were generated using BEAST 1.10.4 [27]. Posterior distributions of the parameters were estimated using Markov chain Monte Carlo sampling, with samples collected every 10,000 steps over 1 × 10^8^ steps. The MCMC analysis was run for more than twice the required length to assess convergence. After discarding the first 25% of samples as burn-in, we verified sufficient sampling by ensuring that the effective sample size (ESS) of each parameter exceeded 200.

## 3. Results

### 3.1. Genetic Diversity Estimates of CTLV Populations

A total of 118 haplotypes were identified among the 181 CTLV isolates, with a haplotype diversity of 98.4% (Table 1). The overall mean nucleotide sequence diversity of these isolates was 0.076 (Table 1). When populations were defined by geographic origin, the highest nucleotide diversity (0.077 ± 0.002) was observed in viral isolates collected from Taizhou City, while the lowest (0.049 ± 0.011) was found in Lishui City. When populations were categorized by host species, the highest nucleotide diversity (0.091 ± 0.046) occurred in viral isolates collected from the orange population, whereas the lowest (0.024 ± 0.009) was detected in the Criton population. Haplotype diversity and nucleotide diversity exceeded 0.500 and 0.005, across all geographic regions and host species, indicating high genetic diversity among the CTLV populations.

To place the genetic diversity of CTLV isolates from Zhejiang within a global evolutionary context, we reconstructed a phylogenetic tree using the 181 CP sequences from Zhejiang alongside 88 available CP sequences retrieved from GenBank. As shown in Appendix A, CTLV isolates can be classified into multiple distinct clades. Remarkably, Zhejiang isolates were distributed across all major clades, indicating that the regional CTLV population encompasses the full spectrum of global phylogenetic diversity and may reflect multiple introduction events or long-term circulation within the region.

### 3.2. Genetic Differentiation Between CTLV Populations

Pairwise *F*_ST_ values for geographic groups of CTLV isolates are shown in Figure 2. Significant genetic differentiation was observed between the Jinhua population and other regions, suggesting a strong spatial structure of the pathogen. Among the comparisons, 26 of 28 *F*_ST_ values were lower than 0.33, indicating a moderate degree of gene flow between these CTLV populations, despite significant population differentiation. Similarly, significant genetic differentiation was found between isolates from the host species Criton and those from other host species, pointing to host-associated genetic structuring, with Criton-derived isolates exhibiting pronounced divergence. In this case, 25 out of 28 *F*_ST_ values were below 0.33, suggesting some degree of gene flow between populations. The results of sliding-window analysis of pairwise *G*_ST_ values for population differentiation among geographic regions and host species are shown in Appendix A. The mean *G*_ST_ values calculated for host species groupings were significantly higher than those for geographic groupings, suggesting that CTLV diversification is influenced more by host species than by geographic factors.

The DAPC analysis revealed genetic differentiation patterns consistent with those observed in the FST analysis. When populations were categorized by geographic region, the DAPC scatterplot classified CTLV isolates into three distinct clusters (Figure 3a). Cluster 1 and Cluster 2 consisted exclusively of viral isolates from Zhoushan and Jinhua, respectively, while Cluster 3 comprised CTLV isolates from multiple regions. When populations were categorized by host species, DAPC classified CTLV isolates into two distinct clusters (Figure 3b). Cluster 1 contained only CTLV isolates from Criton, whereas Cluster 2 included CTLV isolates from multiple host species. These results suggest that both geographic origin and host species contribute to CTLV population differentiation.

### 3.3. Adaptative Evolution of CTLV

When geographic regions were used as grouping factors, phylogeny–trait association analysis revealed no significant signal between geographic origin and phylogenetic relationships, with the exception of isolates from Jinhua, Taizhou and Zhoushan (Table 2, *P*_MC_ < 0.05). However, when the CTLV isolates were clustered by their host species, significant associations were observed between host species and phylogenetic clustering, with the exception of viral isolates from the host species of the pummelo and tangerine (*P*_MC_ > 0.05, Table 2). The BaTS results indicated extensive host-associated variability in CTLV populations, suggesting that host-driven adaptation may be an important evolutionary determinant for CTLV.

The selection analyses revealed that the most polymorphic sites of the CTLV CP gene were under strong purifying selection (Appendix A). Nevertheless, five sites subjected to positive selection were identified in the CTLV CP gene using the codon substitution models implemented in CodeML (Table 3). Among these, only three codons (positions 27th, 38th and 47th) were confirmed by MEME analysis to be undergoing positive selection (Table 3). Positive selection at these sites contributes to increased amino acid variation. Interestingly, the extent of this variation varies among geographic regions (Appendix A). For example, in Taizhou, alanine predominates at the 27th amino acid position, accounting for over 81% of the sequences, whereas in Lishui, alanine accounts for less than 28%, while glutamic acid exceeds 72%. Nonetheless, constrained by limited sample sizes in most regions, further statistical analysis of these differences remains unfeasible at present.

### 3.4. Demographic History of CTLV Population

Mismatch distributions and neutrality tests showed recent historical demographic events, indicated by significant deviations in Fu’s *F*_S_ from neutrality (*p* = 0.018). The hypothesis of sudden expansion was not rejected by analyses of the mismatch distribution, as both Harpending’s raggedness index (*r*) and the sum of square deviations (SSD) were insignificant (*P*_r_ = 0.570 and *P*_SSD_ = 0.30, Figure 4a). Further reconstruction of the demographic history using a Bayesian skyline plot revealed that the size of the CTLV population has fluctuated over time (Figure 4b). Indeed, the population experienced a slight expansion before entering a prolonged period of stability, followed by a recent decline up to the latest sampling year.

## 4. Discussion

Population genetic diversity serves as a key indicator of a pathogen’s capacity to adapt to environmental changes. Pathogens with high genetic diversity are typically better equipped to survive and evolve, enabling them to adapt to new hosts and fluctuating environmental conditions [28]. In line with the characteristics of typical RNA viruses, our study confirmed high genetic diversity among CTLV isolates from different geographic regions and host species (Table 1). However, unlike in other RNA viruses that have recombination hotspots [29], our analysis of the CTLV CP gene revealed no detectable recombination events (phi test, *p* = 0.510). This observation may be explained by strong selective pressures against the survival of novel CTLV recombinants. Supporting this hypothesis, we found the CTLV CPs evolved under exceptionally strong purifying selection (Appendix A), indicating that most of the mutations in this gene were deleterious and consequently eliminated by natural selection.

Within the virus–host–environment interactions, plant viruses are subject to dual selective pressures from both their hosts and the surrounding environment. Consequently, plant RNA viruses have evolved varying degrees of regional or host-specific adaptations [11,30,31]. Supporting this concept, recent analyses of AGSV CP gene sequences revealed distinct nucleotide and protein patterns between apple- and pear-infecting isolates, demonstrating clear host adaptation independent of geographic influences [32]. Our study expands upon these findings by revealing a similar host-associated evolutionary pattern emerge when examining a broader range of host species (Table 2).

The presence of positively selected sites in a gene typically indicates that host-associated selection plays an important role in viral evolution, potentially affecting host range (as exemplified by the P1 gene in potyviruses) [33]. In this study, three positively selected amino acids in the 5′-teminus of the CP gene were identified using both CodeML and MEME analyses (Table 3). These sites appear to contribute to viral evolution, although their specific functional roles remain to be elucidated in future studies.

## 5. Conclusions

In conclusion, this study contributes to our understanding of CTLV genetic diversity at a regional scale. Our findings revealed a high level of genetic variability among CTLV isolates in Zhejiang Province, with significant genetic differentiation influenced by both host species and geographic origin. These findings highlight the importance of local ecological and evolutionary factors in shaping CTLV population dynamics. Future research, particularly phylodynamic analyses involving larger, temporally structured datasets, will offer deeper insights into the evolutionary history of CTLV.

## Figures and Tables

**Figure 1 viruses-17-00909-f001:**
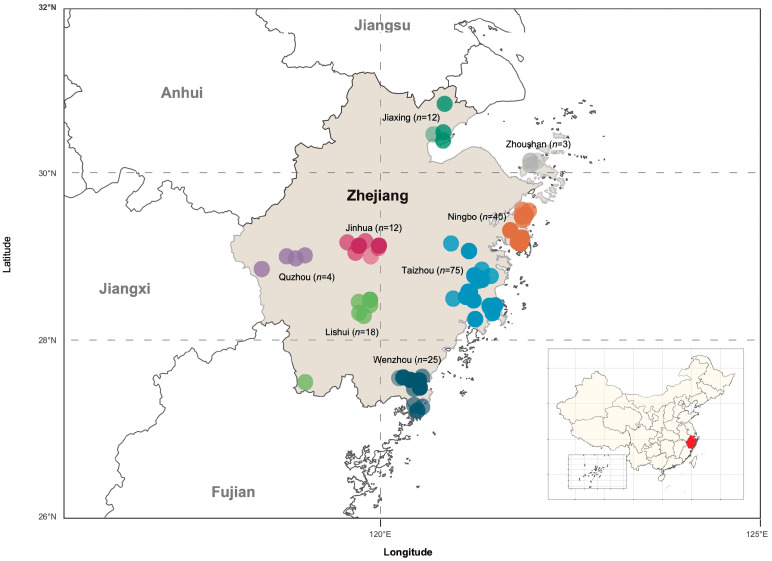
Map showing the localities of citrus tatter leaf virus (CTLV) isolates used in this study. CTLV isolates from different regions are indicated by a unique color. Darker shades indicate regions containing more overlapping viral isolates.

**Figure 2 viruses-17-00909-f002:**
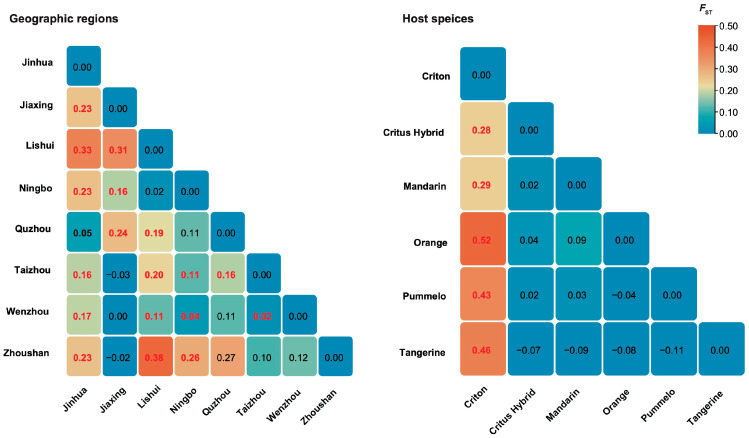
Heatmap showing pairwise *F*_ST_ between CTLV populations. Significant *F*_ST_ values are highlighted in bold.

**Figure 3 viruses-17-00909-f003:**
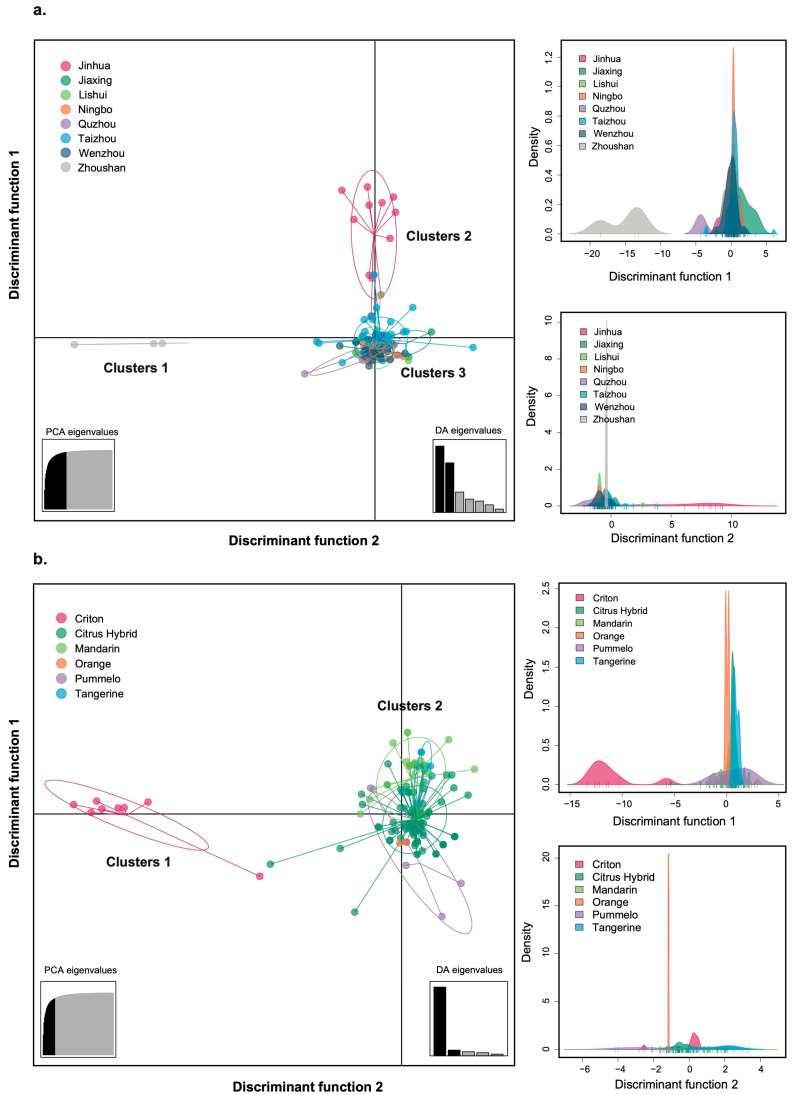
Scatterplots from discriminant analysis of principal components (DAPC) showing predefined CTLV subpopulations by (**a**) geographic region and (**b**) host species. Individual isolates are represented as dots, with the majority contained within inertia ellipses. PCA and DA eigenvalues are displayed in the inset panel. Corresponding discriminant functions for cluster separation are presented on the right panel.

**Figure 4 viruses-17-00909-f004:**
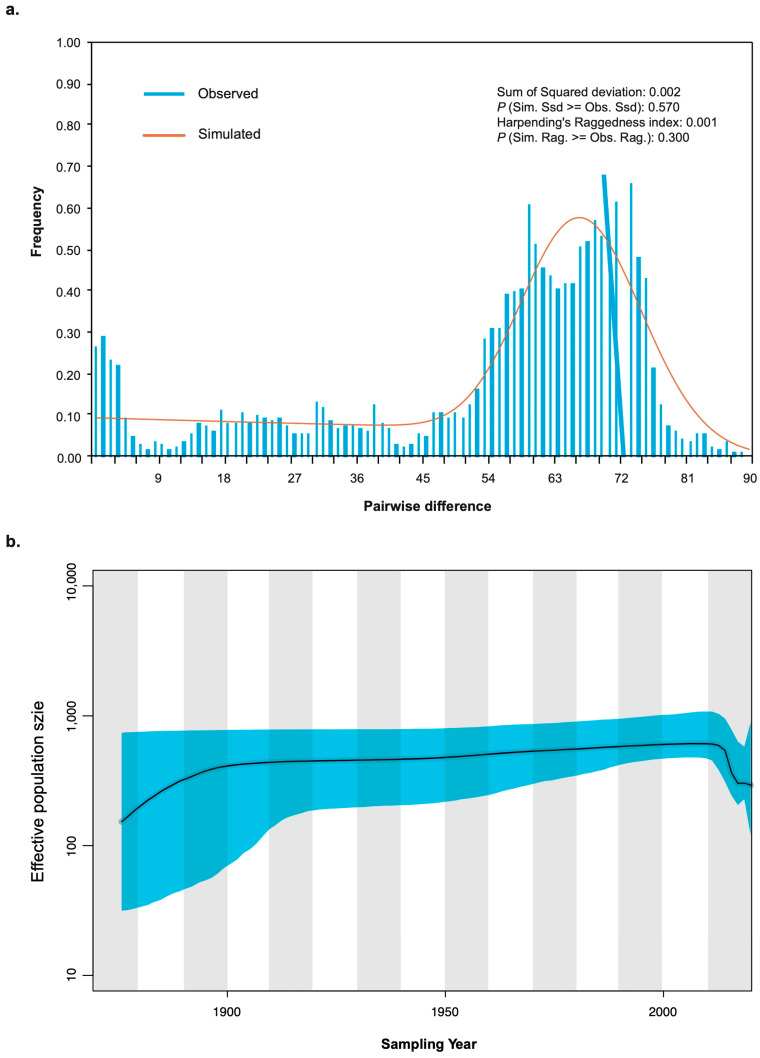
Demographic history dynamics of CTLV population inferred from (**a**) mismatch distribution analysis. (**b**) Bayesian skyline plot. The y axis represents effective population size (N_e_), and the x axis shows calendar years. The black line shows the median population size estimate, with the light blue shading representing the 95% credibility interval.

**Table 1 viruses-17-00909-t001:** Genetic diversity parameters estimated for the CTLV population.

Population		*n*	*s*	*h*	*H* _d_	π	Identity (%)
Region							
	Jinhua	12	125	12.000	1.000 ± 0.034	0.058 ± 0.011	89.36–99.86
	Jiaxing	4	94	4.000	1.000 ± 0.177	0.072 ± 0.018	89.36–99.86
	Lishui	18	121	9.000	0.804 ± 0.091	0.049 ± 0.011	89.36–100
	Ningbo	40	151	23.000	0.928 ± 0.031	0.065 ± 0.005	89.08–100
	Quzhou	4	63	4.000	1.000 ± 0.177	0.054 ± 0.014	92.02–99.72
	Taizhou	75	198	48.000	0.981 ± 0.006	0.077 ± 0.002	87.82–100
	Wenzhou	25	158	23.000	0.993 ± 0.013	0.073 ± 0.003	89.36–100
	Zhoushan	3	57	3.000	1.000 ± 0.272	0.054 ± 0.024	92.02–99.72
Host							
	Criton	8	53	8.000	1.000 ± 0.063	0.024 ± 0.009	93.42–99.86
	Citrus hybrid	138	214	86.000	0.976 ± 0.008	0.075 ± 0.001	87.68–100
	Orange	2	65	2.000	1.000 ± 0.500	0.091 ± 0.046	90.09
	Pummelo	4	108	4.000	1.000 ± 0.177	0.083 ± 0.016	89.78–93.70
	Tangerine	3	90	3.000	1.000 ± 0.272	0.086 ± 0.024	90.76–92.44
Total		181	231	118.000	0.984 ± 0.005	0.076 ± 0.001	

*n*, sample size; *s*, the number of segregation sites; *h*, haplotypes; *H*_d_, haplotype diversity; *π*, nucleotide diversity.

**Table 2 viruses-17-00909-t002:** Results of Bayesian Tip-association significance testing for the geographical and host species on the genetic diversity of CTLV.

Statistic	Observed Mean (95% HPD)	Null Mean (95% HPD)	*p*-Value
Region			
*AI*	7.77 (6.68–8.89)	15.49 (14.52–16.51)	<0.001 ***
*PS*	64.26 (61.00–67.00)	95.30 (92.02–98.40)	<0.001 ***
*MC* (Jinhua)	6.97 (7.00–7.00)	1.25 (1.00–2.00)	0.010 **
*MC* (Jiaxing)	1.00 (1.00–1.00)	1.02 (1.00–1.04)	1.000 ^ns^
MC (Lishui)	2.54 (2.00–4.00)	1.43 (1.00–2.04)	0.110 ^ns^
MC (Ningbo)	3.31 (3.00–5.00)	2.28 (1.89–3.01)	0.100 ^ns^
MC (Quzhou)	1.01 (1.00–1.00)	1.02 (1.00–1.07)	1.000 ^ns^
MC (Taizhou)	7.13 (7.00–8.00)	3.60 (2.88–5.09)	0.010 **
MC (Wenzhou)	2.22 (2.00–3.00)	1.83 (1.17–2.35)	0.440 ^ns^
MC (Zhoushan)	1.99 (2.00–2.00)	1.00 (1.00–1.01)	0.010 **
Host			
*AI*	3.97 (3.21–4.75)	8.06 (7.24–8.97)	<0.001 ***
*PS*	32.13 (30.00–34.00)	41.30 (39.58–42.64)	<0.001 ***
*MC* (Criton)	6.94 (7.00–7.00)	1.08 (1.00–1.31)	0.010 **
*MC* (Citrus hybrid)	25.13 (11.00–31.00)	8.55 (6.57–10.78)	0.020 *
*MC* (Mandarin)	3.01 (3.00–3.00)	1.82 (1.14–2.61)	0.020 *
*MC* (Orange)	n/a	n/a	n/a
*MC* (Pummelo)	1.00 (1.00–1.00)	1.04 (1.00–1.12)	1.000 ^ns^
*MC* (Tangerine)	1.00 (1.00–1.00)	1.01 (1.00–1.00)	1.000 ^ns^

*AI*, association index; *PS*, parsimony score; *MC*, maximum monophyletic clade; HPD, highest probability density interval; n/a: no data available because of insufficient sample size (*n* < 2). Significance thresholds: * 0.01 < *p* < 0.05; ** 0.001 < *p* < 0.01; *** *p* < 0.001; ns, not significant.

**Table 3 viruses-17-00909-t003:** Positively selected sites detected in CP of CTLV using (a) CodeML and (b) MEME.

Model	*np*	*ln* L	2 *Δl* = 2 × (*ln* L1 − *ln* L2)		LRT *p*-Value	Positively Selected Sites (BEB: Pr (*ω* > 1)> 0.5) [ *ω*_ML_]
(a)						
M1a	362	−5349.129				Not allowed
M2a	364	−5349.129	(M1a vs. M2a) 0.000		1.000 ^ns^	[]
M7	362	−5361.708				Not allowed
M8	364	−5344.074	(M7 vs. M8) 35.267		<0.001 ***	27 [1.450], 38 [1.399], 47 [1.364], 103 [1.200], 137 [1.342]
Site	*α*	*β* ^−^	*β* ^+^	LRT	Substitution	
					From	To
(b)						
27	1.94	0.00	45.71	0.050	GCA(Ala)	GAA (Glu), GGA (Gly), ACA(Thr)
38	6.48	0.00	86.08	0.010	GGA (Gly)	AAA(Lys), AGA(Arg), AGC(Ser), AGT(Ser), TCA(Ser)
47	0.00	0.00	7.80	0.010	GGC (Gly)	AGC (Ser), GAC (Asp)
106	0.00	0.00	5.32	0.030	GCC(Ala)	GTC(Val), TCC(Ser)

*np*, number of parameters; *α*, synonymous substitution rate; *β*^−^, non-synonymous substitution rate for the negative/neutral evolution component; *β^+^*, non-synonymous substitution rate for the positive/neutral evolution component; LRT, likelihood ratio test. Significance thresholds: *** *p* < 0.001; ns, not significant.

## Data Availability

The datasets generated for this study are available in the GenBank database, with accession numbers PV567421-PV567502.

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
