# Peer review of "Population Genetic Structure of Citrus Tatter Leaf Virus in Zhejiang Province, China"

_viruses, 2025, doi:10.3390/v17070909_

Round 1
Reviewer 1 Report
Comments and Suggestions for Authors
In the submitted manuscript Lu et al describe the population structure of citrus tatter leaf virus in China primarily using bioinformatics and statistical analysis of 181 coat protein gene sequences. The manuscript does offer insights into CTLVs population structure and host adaptation. However, the manuscript is very heavy on statistics, and it is not clear that all comparisons that could have been made have been. The discussion is very short, which is illustrative of the difficulties in drawing conclusions from such complex statistical analysis. Further analysis of the specific regions undergoing positive selection, identification of domains containing these regions if possible, or expansion of analysis to include other populations outside of China would improve this manuscript greatly. Other issues with language and english use can be improved.
Major issues
1) Where is the dN/dS data? Fig S2 this is one of the only figures cited in the discussion, and warrants inclusion in the main paper. Expansion of dN/dS analysis to include subsets of populations within China, or to include populations outside of China would be very helpful.
2) Supplemental figure labelling and legends can be improved.
3) Missing information from materials ln 98 how was it ligated? What plasmid? ln 101 sequenced using M13 primers??
Minor issues
Ln 50 "holds key position" suggest "is an important proudction region
Citations needed for this sentence and next.
ln 64 type or missing period.
ln 56 suggest "10 varieties cultiviated in Huangyan"
ln 58 should be "grown in "
70 "this study"
Figure 1 What does the inset show? What does the inset within the inset show? The grey circles are hard to see.
Ln 155 what is Orange? Why is it in italics?
Table 1 Any statistical significance to the genomic diversity? Should also include % identity within groups.
ln 175 Citron? Not well defined. I am unfamiliar with this term. More explanation in Intro.
ln 189 Exclusively? Missing words?
ln 192 typo? spacing.
Comments on the Quality of English Language
Some issues were detected and listed above. This was no a rigorous review of english, and suggest authors revise the whole paper with this in mind.
Author Response
In the submitted manuscript Lu et al describe the population structure of citrus tatter leaf virus in China primarily using bioinformatics and statistical analysis of 181 coat protein gene sequences. The manuscript does offer insights into CTLVs population structure and host adaptation. However, the manuscript is very heavy on statistics, and it is not clear that all comparisons that could have been made have been. The discussion is very short, which is illustrative of the difficulties in drawing conclusions from such complex statistical analysis. Further analysis of the specific regions undergoing positive selection, identification of domains containing these regions if possible, or expansion of analysis to include other populations outside of China would improve this manuscript greatly. Other issues with language and English use can be improved.
RESPONSE: We sincerely appreciate the reviewer’s thoughtful evaluation of our manuscript and the constructive suggestions for improvement. As suggested, we have incorporated comparative analysis of global CTLV sequences to complement our focus on Chinese isolates. The discussion has also been expanded to address the evolutionary and practical implications of our findings, particularly in the context of host adaptation and viral evolution. Besides, we have carefully edited the manuscript to improve clarity and readability, ensuring that the language and presentation meet the journal’s standards. We believe these revisions strengthen the manuscript’s contribution to the field.
Major issues
1) Where is the dN/dS data? Fig S2 this is one of the only figures cited in the discussion, and warrants inclusion in the main paper. Expansion of dN/dS analysis to include subsets of populations within China, or to include populations outside of China would be very helpful.
RESPONSE: The dN/dS data presented in Fig. S2 were extracted from the M3 model results generated by CodeML-based method, as descripted our original manuscript. The M3 model provides the posterior probability that each site in the alignment evolved under the different site classes, allowing us to assess selective pressures across the viral genome. Additionally, as suggested by the reviewer, we have expanded our discussion by incorporating a comparative analysis of global CTLV sequences. This complements our primary focus on Chinese isolates and provides broader evolutionary insights into CTLV genetic diversity and selection patterns.
2) Supplemental figure labelling and legends can be improved.
RESPONSE: Thank the reviewer for pointing this out. We have improved the supplemental figures by revising their labels and legends for greater clarity and consistency.
3) Missing information from materials ln 98 how was it ligated? What plasmid? ln 101 sequenced using M13 primers??
RESPONSE: We thank the reviewer for these helpful suggestions. We have revised the text to specify the T-tailed pEASY-T5 cloning vector (TransGen) and M13 primers used for sequencing, providing complete methodological details as suggested.
Minor issues
Ln 50 "holds key position" suggest "is an important production region
RESPONSE: Revised as suggested.
Citations needed for this sentence and next.
RESPONSE: Revised as suggested.
ln 64 type or missing period.
RESPONSE: Revised as suggested.
ln 56 suggest "10 varieties cultivated in Huangyan"
RESPONSE: Revised as suggested.
ln 58 should be "grown in "
RESPONSE: Revised as suggested.
70 "this study"
RESPONSE: Revised as suggested
Figure 1 What does the inset show? What does the inset within the inset show? The grey circles are hard to see.
RESPONSE: Thank the reviewer for pointing this out. In the revised manuscript, we have now moved the phylogenetic tree from Figure 1B to a new supplementary Fig. S1 as suggested by the reviewer #2.
Ln 155 what is Orange? Why is it in italics?
RESPONSE: In the revised manuscript, we have added a concise definition of ‘Orange’ in the M&M section.
Table 1 Any statistical significance to the genomic diversity? Should also include % identity within groups.
RESPONSE: Thank the reviewer for pointing this out. In population genetic analyses, pairwise differentiation tests assess statistical significance between populations, whereas genetic diversity estimates do not require statistical testing. As suggested by the reviewer, we have included the identity within groups in the revised Table 1.
ln 175 Citron? Not well defined. I am unfamiliar with this term. More explanation in Intro.
RESPONSE: We appreciate the reviewer’s comment and agree that additional context would be helpful. In the revised manuscript, we have added a concise definition of ‘Citron’ in the M&M section.
ln 189 Exclusively? Missing words?
RESPONSE: We appreciate the reviewer's careful reading. We have corrected this sentence in the revised manuscript to improve clarity and accuracy.
ln 192 typo? spacing.
RESPONSE: Revised as suggested
Comments on the Quality of English Language
Some issues were detected and listed above. This was no a rigorous review of English, and suggest authors revise the whole paper with this in mind.
RESPONSE: We thank the reviewer for highlighting these language issues. We have carefully reviewed and improved the English throughout the manuscript, with particular attention to the points noted.
Reviewer 2 Report
Comments and Suggestions for Authors
The study by Lianming and co-authors investigates the genetic diversity and evolution of Citrus tatter leaf virus (CTLV) in Zhejiang Province, China, using 181 coat protein gene sequences, 84 of which were newly determined in the frame of this work. The results revealed extensive genetic variation driven by geography and by host species. Selection pressure analysis showed most genetic sites were under purifying selection, while a number of clade-specific codons were revealed to be under positive selection. Global analysis of all available sequences showed that the isolates from Zhejiang were distributed across all major clades, supporting multiple introductions or long-term circulation. The presented Communication overall expands the understanding of the genetic diversity of CTLV, by focusing on the Zhejiang geographic region, stimulating future novel research on a larger scale.
Please find below suggestions and comments for improvement:
L42: choloris
L49: in viral infection
L81: hosts, including
L82: please better specify the term “Criton”; si this a local variety? Of which species?
L82: Citrus should not be capital
L83: Tangerine should not be capital
Figure 1 legend: delete “respectively”; what are the different tones of the colors? For example in Taizhou there are different tones of blue in Jinhua of red, , but the legend does not clarify the reason of that.
Figure 1, panel B: the panel appears few informative as presented; probably it could be better to present as independent (supplementary) figure, with information for each branch regarding host and location,so to show the sequence grouping
L126: verify double minus
L155: orange
L155: were detected in ….
L155: check again “Criton”, see comment above
L161: the number of all publicly available sequences at the time of analysis should be specified
L173: to better guide the reader, please specify why the value of 0.33 is a cut-off value
L192: edit “Citron”
L200: edit sentence, something is missing or needs rewriting “which optimally discriminant between..”
L202: on the right
L205: …revealed no significant…
L209: plant names not capital
L266: bracket is missing
Author Response
The study by Lianming and co-authors investigates the genetic diversity and evolution of Citrus tatter leaf virus (CTLV) in Zhejiang Province, China, using 181 coat protein gene sequences, 84 of which were newly determined in the frame of this work. The results revealed extensive genetic variation driven by geography and by host species. Selection pressure analysis showed most genetic sites were under purifying selection, while a number of clade-specific codons were revealed to be under positive selection. Global analysis of all available sequences showed that the isolates from Zhejiang were distributed across all major clades, supporting multiple introductions or long-term circulation. The presented Communication overall expands the understanding of the genetic diversity of CTLV, by focusing on the Zhejiang geographic region, stimulating future novel research on a larger scale.
RESPONSE: We thank the reviewer for these positive comments.
Please find below suggestions and comments for improvement:
L42: choloris
RESPONSE: Revised as suggested.
L49: in viral infection
RESPONSE: Revised as suggested.
L81: hosts, including
RESPONSE: Revised as suggested.
L82: please better specify the term “Criton”; is this a local variety? Of which species?
RESPONSE: In the revised manuscript, we have added a concise definition of ‘Criton’ in the M&M section.
L82: Citrus should not be capital
RESPONSE: Pease see our response above.
L83: Tangerine should not be capital
RESPONSE: In the revised manuscript, we have added a concise definition of ‘Tangerine’ in the M&M section.
Figure 1 legend: delete “respectively”; what are the different tones of the colors? For example, in Taizhou there are different tones of blue in Jinhua of red, , but the legend does not clarify the reason of that.
RESPONSE: We have removed "respectively" from the figure legend as suggested. The color gradients in the figure represent data density, with darker shades indicating areas of higher viral isolate concentration due to marker overlap. This explanation has been added to the revised legend for clarity.
Figure 1, panel B: the panel appears few informative as presented; probably it could be better to present as independent (supplementary) figure, with information for each branch regarding host and location, so to show the sequence grouping.
RESPONSE: We thank the reviewer for their valuable suggestion. In the revised manuscript, we have moved the phylogenetic tree from Figure 1B to a new Supplementary Fig. S1. This standalone figure now includes comprehensive host and location annotations for each branch, allowing for clearer visualization of sequence groupings while maintaining the narrative flow of the main text.
L126: verify double minus
RESPONSE: Revised as suggested.
L155: orange
RESPONSE: Revised as suggested.
L155: were detected in ….
RESPONSE: Revised as suggested.
L155: check again “Criton”, see comment above
RESPONSE: Changed as suggested. Pease see our response above.
L161: the number of all publicly available sequences at the time of analysis should be specified
RESPONSE: Specified as suggested.
L173: to better guide the reader, please specify why the value of 0.33 is a cut-off value
RESPONSE: We appreciate this helpful suggestion. In the revised manuscript, we have added a brief explanation for the FST threshold of 0.33 in the M&M section.
L192: edit “Citron”
RESPONSE: Revised as suggested.
L200: edit sentence, something is missing or needs rewriting “which optimally discriminant between..”
RESPONSE: We have rephrased the figure legend in the revised Figure 2.
L202: on the right
RESPONSE: Revised as suggested.
L205: …revealed no significant…
RESPONSE: Revised as suggested.
L209: plant names not capital
RESPONSE: Revised as suggested.
L266: bracket is missing
RESPONSE: Revised as suggested.
Round 2
Reviewer 1 Report
Comments and Suggestions for Authors
The authors have addressed my concerns